# Analysis of Vibration Monitoring Data of Flexible Suspension Lifting Structure Based on Time-Varying Theory

**DOI:** 10.3390/s20226586

**Published:** 2020-11-18

**Authors:** Qifeng Peng, Peng Xu, Hong Yuan, Haixia Ma, Jianghong Xue, Zhenya He, Shanqing Li

**Affiliations:** 1MOE Key Laboratory of Disaster Forecast and Control in Engineering, School of Mechanics and Construction Engineering, Jinan University, Guangzhou 510632, China; pengqifeng@stu2016.jnu.edu.cn (Q.P.); xupeng@stu2019.jnu.edu.cn (P.X.); txuej@jnu.edu.cn (J.X.); lishanqing2012@jnu.edu.cn (S.L.); 2School of Electrical Engineering, South China University of Technology Guangzhou College, Guangzhou 510800, China; mahx@gcu.edu.cn; 3Guangdong Provincial Key Laboratory of Precision Equipment and Manufacturing Technology, South China University of Technology, Guangzhou 510640, China; mezhyhe@scut.edu.cn

**Keywords:** machine dynamics, non-linear vibration, random vibration, sensors and actuators, structural dynamics and control

## Abstract

An elevator is a typical flexible lifting machine. In order to monitor the vibration of elevator structure, the vibration characteristics of an elevator with a traction ratio of 1:1 has been tested experimentally. Sensors were arranged on the platform frame, car roof, and hoist rope to test the vibrations of the elevator in both ascending and descending conditions. The transverse, longitudinal, and coupled transverse-longitudinal vibrations were compared and analyzed. Further, the short-time Fourier transform (STFT) method was used to examine the power spectral density (PSD) of the test results, and the main frequency distribution and influencing factors of the vibration of elevator components were investigated. The results revealed that the transverse and longitudinal vibrations of the platform frame were low-frequency vibrations, which was attributed to the elastic interaction between the platform frame and the car frame. The form and amplitude of longitudinal vibration of the car frame were basically consistent with those of the platform frame, but PSD of the transverse vibration had an obvious peak in the high-frequency region. The transverse and longitudinal vibration frequencies of the hoist rope were higher. Furthermore, the peak PSD value of transverse vibration of the hoist rope was 421 times larger than that of the car frame, so a small disturbance at the end of the rope could lead to a huge disturbance in the center of the rope. Overall, this study provides useful insights on designing an elevator monitoring sensor and relevant data processing.

## 1. Introduction

With the rapid growth of the elevator industry globally, the safety and reliability requirements of elevators are continuously increasing. Although the safety level of elevators continues to improve with the advancements in research, safety risks still exist due to their extensive utilization, and the safety performance of elevators needs to be enhanced. Further, elevator accidents have aroused widespread concern in society. Such accidents not only cause casualties and property losses but also have a bad social impact. Improper handling may even cause panic and affect social stability. Therefore, elevator safety has become an important part of public security.

Elevator performance analysis based on vibration monitoring as well as the number and location of sensors during monitoring has garnered considerable research attention. An appropriate selection of the number and location of monitoring points and to accurately determine the operation status of the structure with minimum number of sensors is a direction worthy of research. Due to the continuous changes in the length of wire rope during the operation, the dynamic characteristics of the elevator at different positions are different [1,2,3,4]. Therefore, it is crucial to investigate the vibration characteristics of key components such as elevator tractor, hoist rope, car frame, and platform frame. Furthermore, after obtaining the monitoring data, the analysis of monitoring data, determining the maximum value of the main frequency and vibration amplitude during structural operation, and assessing the operation status and safety of the structure, have a high application significance for safety performance evaluation of elevator structure.

Zhang et al. [5] introduced a novel and convenient method for monitoring the comfort of elevator rides using smart phones. Watanabe et al. [6] studied the vertical vibration of a multi-degree-of-freedom elevator traction system with compensation rope and suggested that the maximum vertical vibration of the system occurred when the elevator was decelerating. Further, they analyzed the influence of braking torque on the vertical displacement of the compensation wheel. Jovan et al. [7] simulated the dynamic characteristics of elevators and investigated the influence of dynamic parameters such as velocity and acceleration, lifting height, rated load, and cabin weight on these characteristics. Gaiko et al. [8] considered the hoist rope as a beam of variable length with constant axial vertical velocity to examine its transverse vibration under wind bracing using singular perturbation theory and verified the results by numerical analysis. Fan et al. [9] developed a model for the hoist rope using non-singular beam and analyzed its natural frequency and vibration pattern after the car position changed. Further, they compared the results with the Abaqus analysis results. Zhu et al. [10] explored the dynamic characteristics of hoist rope as well as the free vibration and forced vibration characteristics of the car. Khazanovich et al. [11] established the dynamic equations of a system composed of traction machine, hoist rope, car, and counterweight under starting, steady-state motion, and braking conditions, and proposed a relation for calculating the equivalent load of the elevator power unit. Esteban et al. [12] proposed a dynamic model for calculating the dynamic variation in acceleration of the elevator under normal operation, and compared the simulation results with the experimental results.

These studies provided a vital reference for examining the vibration characteristics of elevator traction systems. However, most of them focused on the vibration theory and simulation study of the hoist rope under operating conditions, and only a few of them used the actual engineering structure to test the vibration characteristics and time-frequency characteristics of the traction system. In this study, an elevator with a traction ratio of 1:1 was used as the experimental platform. Sensors were arranged in the car, on the roof, and on the hoist rope to test the vibration characteristics of the elevator under both ascending and descending conditions. Further, the transverse, longitudinal, and coupled transverse-longitudinal vibration of platform frame, car frame, and hoist rope were analyzed. The time-frequency characteristics of the system were examined by the short-time Fourier transform (STFT) method. Furthermore, the frequency distribution of vibration signals corresponding to different elevator components was investigated under various operating conditions.

## 2. Vibration Characteristics of Traction System

The operation of a conventional elevator is based on the friction between the hoist rope and the traction wheel. Under external excitations, the movement of the system inevitably produces vibrations. These external excitations include the traction force provided by the traction machine, the braking force produced by the brake, the impact force caused by the irregularity of the guide rail, and the air resistance force generated during the operation [13,14,15]. When the elevator structure is considered as a unified dynamic system, its physical characteristics mainly include mass, stiffness, and damping. According to the time dependence of physical properties, the structure can be divided into time-invariant and time-varying structure [16]. Under stationary state or emergency stop condition, since the length of the hoist rope exhibits a minor change, the elevator structure is considered as a time-invariant system. However, during the entire elevator operation, the length of the hoist rope is constantly varying. Consequently, the hoist rope can be considered as an axial moving string [1]. Here, a method to identify the time-varying structural parameters is adopted to analyze the dynamic response of the system, and the system is considered to exhibit linear time-varying characteristics [17,18,19,20,21,22,23]. A linear time-varying system can be represented by a second-order differential equation with variable coefficients,
(1)$$M\left(t\right)\ddot{x}\left(t\right)+C\left(t\right)\dot{x}\left(t\right)+K\left(t\right)x\left(t\right)=f\left(t\right),$$
where f(t) and x(t) are the external excitation and displacement response vectors, respectively. M(t), C(t), and K(t) are the mass matrix, damping matrix, and stiffness matrix of the time-varying structural system, respectively.

The frequency-domain characteristics of mechanical vibrations are mainly expressed in terms of power spectral density (PSD). Since the integral of a random signal may not converge, the Fourier transform of the signal can neither be obtained directly nor can it be accurately described by mathematical expressions. Therefore, a random signal can only be expressed by statistical methods. The PSD function represents the statistical average of spectral characteristics of a signal.

PSD can be used to analyze the strength of a frequency-domain signal in terms of energy. For the power spectrum of simple harmonic vibration, the periodic function x(t) can be written as a Fourier series,
(2)$$x\left(t\right)=a_{0}+\sum_{k=1}^{\infty }\left[\frac{1}{2}\left(a_{k}+jb_{k}\right)e^{-jk\omega_{0}t}\right]=c_{0}+\sum_{k=1}^{\infty }\left(c_{k}e^{jk\omega_{0}t}+c_{k}^{*}e^{-jk\omega_{0}t}\right).$$

We define $c_{0}=a_{0}$, $c_{k}=\frac{1}{2}\left(a_{k}-jb_{k}\right)$, and $c_{k}^{*}=\frac{1}{2}\left(a_{k}+jb_{k}\right)$, when $c_{0}=0$, the *k*-order expansion $c_{k}e^{jk\omega_{0}t}+c_{k}^{*}e^{-jk\omega_{0}t}$ of x(t) can be represented by a rotating vector in complex plane. The components $c_{k}$ and $c_{k}^{*}$ rotate with angular velocities of $k\omega_{0}$ and $-k\omega_{0}$, respectively, and the resultant effect is a real vibration function with an amplitude of $\left|C_{k}\right|=2\;\left|c_{k}\right|$, so Equation (2) can be written as:(3)x(t)=12∑k=1∞(Ckejkω0t+Ck*e−jkω0t)=Re[∑k=1∞Ckejkω0t]

The root mean square (RMS) value of x(t) can be expressed as follows:(4)$$\begin{array}{c} \;\bar{x^{2}\left(t\right)}=\frac{1}{T}\overset{T}{\underset{0}{\int }}x^{2}\left(t\right)dt=\frac{1}{T}\overset{T}{\underset{0}{\int }}\frac{1}{4}\overset{\infty }{\underset{k=1}{\sum }}{\left(C_{k}e^{jk\omega_{0}t}+C_{k}^{*}e^{-jk\omega_{0}t}\right)}^{2}dt\\ =\frac{1}{4}\overset{\infty }{\underset{k=1}{\sum }}\left[\frac{C_{k}^{2}e^{j2k\omega_{0}t}}{j2k\omega_{0}t}+\frac{2C_{k}C_{k}^{*}}{T}t+\frac{{\left(C_{k}^{*}\right)}^{2}e^{-j2k\omega_{0}t}}{-j2k\omega_{0}t}\right]\\ =\frac{1}{2}\overset{\infty }{\underset{k=1}{\sum }}C_{k}C_{k}^{*}=\frac{1}{2}\overset{\infty }{\underset{k=1}{\sum }}{\left|C_{k}\right|}^{2}=\overset{\infty }{\underset{k=1}{\sum }}{\left(\frac{1}{\sqrt{2}}\left|C_{k}\right|\right)}^{2} \end{array}$$
where $\frac{1}{2}C_{k}C_{k}^{*}=\frac{1}{2}{\left|C_{k}\right|}^{2}$ represents the contribution of the *k*^th^-order harmonic component, and $\bar{x^{2}\left(t\right)}$ indicates power. Consequently, $\frac{1}{2}C_{k}C_{k}^{*}$ represents the total power at frequency, and $W\left(f_{k}\right)$ is used to represent the power of the signal in the frequency segment ∆f, which is called the spectral power,
(5)$$W\left(f_{k}\right)=\frac{1}{2}C_{k}C_{k}^{*}$$

Therefore, the RMS value of x(t) can be expressed as the sum of spectral power over all frequencies, which is called the power spectrum and is expressed as follows:(6)$$\bar{x^{2}\left(t\right)}=\overset{\infty }{\underset{k=1}{\sum }}W\left(f_{k}\right)$$

A power spectrum indicates the distribution of the energy of a signal among its different frequency components. The power spectrum divided by the frequency interval, ∆f, is called PSD $G\left(f_{k}\right)$,
(7)$$G\left(f_{k}\right)=\frac{W\left(f_{k}\right)}{∆f}=\frac{C_{k}C_{k}^{*}}{2∆f}$$

Consequently, the RMS value can be represented in terms of PSD as:(8)$$\bar{x^{2}\left(t\right)}=\overset{\infty }{\underset{k=1}{\sum }}G\left(f_{k}\right)∆f$$

The spectral power function is typically used to analyze periodic functions, and the result is in the form of a discrete spectrum, while non-periodic functions and the random functions need to be represented by PSD.

Combined with the actual test results of the elevator in use, the vibration of the car, car frame, and hoist rope are analyzed to determine the best position of the monitoring sensor. The monitoring data is examined and processed by using time-varying theory and frequency spectrum analysis. The relationship between the monitoring data and the running state of the elevator is investigated (as shown in Figure 1). In the following sections, the specific experimental process and analysis method are described.

## 3. Vibration Testing Methods and Experimental Parameters

In this section, the arrangement of test points, experimental equipment, and methods, which are employed to analyze the vibration characteristics of hoist rope, car frame, and platform frame under ascending and descending operation, are described.

### 3.1. Arrangement of Measuring Point and Direction Definition

#### 3.1.1. Arrangement of Measuring Point

Capacitive acceleration sensors are arranged on the platform frame, car frame, and hoist rope, as shown in Figure 1. It is worth noting that the vibration measurement point of the car roof rope is only used for an elevator with traction ratio of 1:1. For such an elevator, there is no significant relative movement between the wire rope and the car roof, while the hoist rope in an elevator with traction ratio of 2:1 rotates around the guide wheel on the car roof, so the sensor cannot be fixed on the rope.

Since longitudinal vibration is the most crucial vibration parameter, several measurement points were arranged in the Z direction. A total of six longitudinal measurement points, five transverse measurement points, and one triaxial measurement point (point 3) were considered in the test. The specific arrangement and direction of measurement points are shown in Table 1.

#### 3.1.2. Direction Convention

The directions parallel to and perpendicular to the elevator door are considered as X and Y directions respectively, and the vertical direction is set as Z direction, as shown in Figure 2. The transverse vibration can be divided into two directions: X and Y directions. The longitudinal vibration occurs in the Z direction.

### 3.2. Test Equipment and Methods

To realize a realistic experiment, an in-service elevator with traction ratio of 1:1 and lifting height of 7.8 m was considered as the experimental object. A no-load test was conducted. The specific parameters of the tested elevator are shown in Table 2.

The dynamic characteristics of the structure were determined by analyzing the vibration signal. The vibration characteristics of the elevator were tested in normal operating condition under both descending and ascending conditions. The experimental setup is shown in Figure 3.

A capacitive sensor is used in this experiment (as shown in Figure 3). It uses change in capacitance to measure the change in voltage. Capacitive sensor has the advantages of simple structure, small volume, high resolution, and good dynamic response. It is especially suitable for the acquisition of elevator acceleration signal data needed in this paper. The parameters of the capacitive accelerometer are shown in Table 3.

According to the actual structural characteristics of the elevator, sensors are arranged, and the vibration characteristics of different parts are analyzed. The running state of the elevator is monitored in real-time. This method can measure the running state of elevators stably and reflect the vibration characteristics of different parts of elevators in real-time. To compare the vibration characteristics of different elevator structures during normal operation, the actual monitoring data of an elevator is analyzed in the following section.

## 4. Analysis of Test Results for Normal Descending Operation of Elevator without Load

### 4.1. Vibration Characteristics of Platform Frame

The platform frame is used to load passengers, and its vibration characteristics directly affect the ride comfort. In this experiment, four acceleration sensors were arranged in the car to test the vibration of the cage in X, Y, and Z directions. Figure 4 shows the arrangement of sensors in the experimental cage.

#### 4.1.1. Transverse and Longitudinal Vibration of Platform Frame

The experimentally measured horizontal and vertical vibration acceleration of the elevator car body under normal descending state with no load are shown in Figure 5. Here, the operation of the elevator is divided into three stages according to its running state: acceleration period, constant speed period, and deceleration period (these stages are divided by red dotted lines in Figure 5).

The transverse vibration of the platform frame in the X direction can be characterized as follows. During the acceleration period of the elevator, the vibration amplitude of the platform frame basically fluctuates around 0. The maximum vibration acceleration is nearly $\;0.05\;m/s^{2}$. During the constant speed period, the amplitude of vibration increases, and the maximum acceleration is $0.42\;m/s^{2}$. During the deceleration period, the vibration is restored to the vicinity of 0.

The transverse vibration of the platform frame in the Y direction can be characterized as follows. The vibration in the Y direction during the acceleration period is more obvious than that in the X direction, but the maximum vibration acceleration is less than 0.05$\;m/s^{2}$. During the first 1.5 s of the constant speed period, the vibration acceleration changes between $-0.05\;\mathrm{and}\;0.05\;m/s^{2}$. There is a noticeable vibration in the constant speed period, and the vibration acceleration varies from $-0.11\;\mathrm{to}\;0.19\;m/s^{2}$. The vibration acceleration fluctuates between 0  and $0.1\;m/s^{2}$ during the deceleration period.

The longitudinal vibration of the platform frame in the Z direction can be characterized as follows. Z direction is considered as the operating direction of the elevator, so any change in the speed of the elevator is directly reflected in the acceleration curve of the Z direction. It can be seen from the vibration curve in Figure 5 that during the acceleration period, the longitudinal vibration acceleration of the platform frame initially increases and then decreases, and the maximum vertical acceleration is $0.72\;m/s^{2}$. The longitudinal vibration during the constant speed period is random, and the vibration acceleration is no more than $0.1\;m/s^{2}$. During the deceleration period, the absolute value of the longitudinal deceleration of the platform frame initially increases and then decreases, and the absolute value of the maximum longitudinal deceleration is $0.72\;m/s^{2}$.

It is clear from the experimental results that when the elevator is normally descending, the main vibration of the platform frame is longitudinal vibration, and the maximum acceleration of the longitudinal vibration is 3.79 times that of the transverse vibration. The transverse vibration of the platform frame is not obvious during the acceleration period of the elevator, but it increases during the constant speed period and decreases during the deceleration period. At the end of the deceleration, the longitudinal and transverse vibrations of the platform frame are stopped.

#### 4.1.2. Time-Frequency Characteristics of Transverse and Longitudinal Vibrations of Platform Frame

During the operation period, the position of the elevator car is constantly changing, which in turn changes the length of the wire rope from the top of the car to the contact point of the traction wheel. Therefore, the entire operation process is nonlinear and unstable, and the discrete statistical features of vibration are time-varying. Here, Hanning window was used for the spectral analysis of the data, and the STFT method was used for the time-frequency analysis of the vibration signals. The power spectrum and time-frequency characteristic curve of the data were obtained (the same analysis method was employed for the car frame and the hoist rope). The three-dimensional time-frequency characteristics of vibration signals of the platform frame in different directions are shown in Figure 6. The variation in the vibration frequency and vibration intensity of the platform frame with time is evident in Figure 6.

It can be seen from the power spectrum analysis results that the vibration frequencies of the platform frame in X, Y, and Z directions are all below 10 Hz, which means that the transverse and longitudinal vibrations mainly occur at low frequencies. The peak value of PSD for the platform frame is 0.045 m^2^/s^3^ in the X direction, 0.0019 m^2^/s^3^ in the Y direction, and 0.17 m^2^/s^3^ in the Z direction. This indicates that the vibration energy of the platform frame is minimum in the Y direction. The vibration energy in the X direction is 23.7 times higher than that in the Y direction. This implies that during normal operation, the transverse vibration of the platform frame is mainly concentrated in the X direction, which is caused by the unevenness of the guide rail. The vibration energy in the Z direction is the largest, which is 89 times higher than that in the Y direction and 3.8 times higher than that in the X direction, indicating that the vibration energy is primarily concentrated in the Z direction.

The time-domain frequency spectrum corresponding to the location of peak value in the time-frequency characteristic curve is defined as the main frequency component of the vibration. The peak frequency in the main frequency component is defined as the main frequency. By analyzing the time-frequency characteristic spectrum for the three directions, it can be found that the main frequency of X, Y, and Z directions is around 1 Hz, indicating that the main frequency of the three-way vibration of the elevator platform frame is similar. This can be attributed to the fact that there is a rubber pad between the platform frame and the car frame. The gravity of the entire platform frame basically acts on the rubber pad. The rubber pad can be considered as an isotropic elastomer, so the platform frame shows similar vibration frequencies in the three directions. From the perspective of time dimension, all the vibration energy peaks in the X and Y directions appear during the constant speed stage in the middle of the operation time period, indicating that the lateral vibration of the platform frame is related to the position of the car. The vibration energy peaks in the Z direction appear during the acceleration and deceleration periods, indicating that the longitudinal vibration characteristics of the platform frame are related to the running state of the elevator.

### 4.2. Vibration Characteristics of Car Frame

The car frame is the load-bearing component of the car, and the weight of the platform frame and the load inside the car are directly transferred from the car frame to the hoist rope. Therefore, the car frame needs to have sufficient strength and stiffness to meet the normal operation requirements of the elevator. The experimental layout of sensing equipment on the car roof is shown in Figure 4.

#### 4.2.1. Transverse and Longitudinal Vibration of Car Frame

The transverse and longitudinal vibration curves of the elevator car frame are shown in Figure 7. The transverse vibration of the car frame in the Y direction can be explained as follows. The vibration amplitude increases with the speed during the acceleration period and reaches the peak value during the later stage of acceleration. The maximum vibration acceleration is $0.09\;m/s^{2}$. During the uniform speed stage, the amplitude of vibration is relatively stable, and the vibration acceleration varies from  −0.14 to $0.12\;m/s^{2}$. The vibration amplitude gradually decreases during the deceleration state.

Similar to the vibration of the platform frame in the Z direction, the vibration of the car frame in the Z direction remains basically stable, and the maximum acceleration is $0.72\;m/s^{2}$. The absolute value of maximum deceleration is $0.70\;m/s^{2}$. It can be seen that both the vibration mode and the vibration amplitude of the car frame are essentially the same as that of the platform frame.

It is clear from the experimental results that similar to the platform frame, the main vibration of the car frame during descending operation is longitudinal vibration. The maximum acceleration of the longitudinal vibration is 5.14 times higher than that of the transverse vibration. In contrast to the platform frame, the transverse vibration of the car frame is slowly attenuated. After the deceleration, the longitudinal vibration of the car frame basically stops, while the transverse vibration does not stop completely.

#### 4.2.2. Time-Frequency Characteristics of Transverse and Longitudinal Vibration of Car Frame

Similar to the platform frame, the vibration of the car frame is time-varying. The three-dimensional time-frequency characteristics of the car’s vibration signals in the Y and Z directions are shown in Figure 8. The variation in the vibration frequency and vibration intensity of the car frame with time is clearly seen in Figure 8.

The peak value of PSD for the car frame in the Z direction is nearly $0.17{\;m}^{2}/s^{3}$, and the main frequency is in the range of 0 to 1 Hz, which is still concentrated in the low-frequency region. A clear difference from the vibration of the platform frame is that the peak value of PSD in the Y direction is nearly $3.5\times 10^{-4}{\text{ m}}^{2}/{\text{s}}^{3}$, which is only 1/5 of the peak value in the same direction for the platform frame. In addition, the main frequency of the platform frame is concentrated around 1 Hz, while that of the car frame is around 12 Hz, and there are obvious peaks around 29 and 46 Hz. This indicates a firm installation of the car frame as well as a reasonable clearance between the guide wheel and the guide rail.

### 4.3. Vibration Characteristics of Hoist Rope

An elevator system comprises of multiple hoist ropes to lift heavy loads. To ensure the safety of elevator equipment and passengers, the strength and number of hoist ropes must be enough to support the maximum load applied. Several studies on the vibration of elevator wire rope considered hoist rope as an entire elastic system, but a comparative analysis of the vibration characteristics of multiple trailing wires has been rarely presented.

Further, in practical engineering, the varying length and mechanical properties of the wire ropes as well as the wear and aging of the traction wheel may lead to inconsistent tension between the wire ropes. Such inconsistency can cause serious wear of the traction pulley in the rope groove where the tension is high and can even lead to serious elevator accidents. In this section, based on practical engineering experiment, the vibration characteristics of hoist rope are analyzed.

#### 4.3.1. Test Point Distribution of Hoist Rope

The elevator considered here has five hoist ropes, and the arrangement of sensors on these ropes is shown in Figure 4. A three-axis sensor is located on rope no. 3 to measure the vibration in the X, Y, and Z directions. Sensors are arranged on ropes no. 2 and no. 5 to measure the Y-direction vibration Sensors are arranged on ropes no. 1 and no. 4 to measure the Z-direction vibration. The height of the sensors is approximately 1 m from the rope end of the car top.

#### 4.3.2. Coupled Transverse-Longitudinal Vibration of Hoist Rope

The hoist rope no. 3 is fitted with a triaxial sensor. Consequently, the vibration of this rope in X, Y, and Z directions is considered to analyze the coupled transverse-longitudinal vibration characteristics, which are shown in Figure 9. It is evident that the initial vibration of the hoist rope in the three directions exhibits some hysteresis. The vibration decays faster in the later stage. Lateral vibration stops before the elevator is completely stopped. The longitudinal vibration acceleration gradually decreases in the late deceleration stage. It is clear from Figure 9 that during the uniform speed stage, the vibrations of hoist rope in the three directions are not independent, and if the vibration in one direction increases, the vibration in the other two directions also increases.

Comparing the transverse vibration in the X and Y directions (Figure 9), it can be seen that the vibration trends are basically the same, and the vibration amplitudes have a minor difference. The peak value of vibration deceleration in the X and Y directions is larger than that of vibration acceleration during the acceleration, uniform speed, and deceleration stages. This is related to the gravity of the sensor. The sensor is fixed on the hoist rope by tape. When the elevator descends at a constant speed, the gravity of the sensor causes the absolute value of the acceleration to be smaller than that under the ascending condition. Therefore, the test results should be corrected according to the weight of the sensor when the vibration characteristics of the hoist rope are analyzed.

#### 4.3.3. Time-Frequency Analysis of Coupled Transverse-Longitudinal Vibration of Hoist Rope

The three-dimensional time-frequency characteristics of vibration signals of hoist rope no. 3 in the X, Y, and Z directions are shown in Figure 10. It can be seen that the time-frequency characteristics of the hoist rope are more obvious than that of platform frame and car frame. The variation in the vibration frequency and vibration intensity of the hoist rope at various times is evident in Figure 10. The peak values of PSD of the hoist rope in X, Y, and Z directions are 1.21, 1.48, and $0.14{\;m}^{2}/s^{3}$, respectively. This indicates that the transverse vibration energy of the hoist rope is much larger than the longitudinal vibration energy. There are three obvious energy peaks around 12, 26, and 40 Hz in both X and Y directions, which suggests that the transverse vibration frequency spectrum of the hoist rope is similar in the three directions. Therefore, any of three directions (X, Y, or Z) can be selected for analyzing the transverse vibration characteristics of the hoist rope. The energy peaks in the Z direction appear near 0 and 29 Hz, and the time-frequency characteristics are obviously different than that for transverse vibration.

### 4.4. Comparison and Discussion

For a comparative analysis, the vibration and frequency characteristics of the elevator under the uniform running stage were analyzed. The vibration acceleration peaks of each component are shown in Table 4.

The above analysis reveals that the overall vibration characteristics of the platform frame and car frame are similar, while the transverse vibration of the hoist rope is the largest. When the elevator is under normal operation, it is subjected to the drag force of the tractor, impact force caused by irregular guide rail, air resistance, etc., and the vibration response is generated. The car body and the car frame show relatively stable performance. Based on the analysis of time-frequency characteristics, it can be inferred that the car frame is stimulated more frequently and can reflect the running state of the traction system more accurately. Therefore, while monitoring the running state of the elevator, the monitoring point should be selected on the load-bearing structure of the car roof.

## 5. Vibration Characteristics of Hoist Rope under Normal Ascending Operation without Load

In this section, the vibration of the hoist rope of the platform frame is analyzed under normal ascending operation, and the main vibration characteristics under the ascending and descending conditions are compared.

A key feature of the elevator traction system is that the weights on both sides of the traction wheel are different. Therefore, the descending operation without load can be regarded as dropping the light load, and the ascending operation without load can be considered as lifting the light load. The vibration of the car frame can be considered as the vibration of the rope tail. The vibration at a distance of 1 m from the car frame is considered as the vibration in the rope.

### 5.1. Comparison of Transverse Vibration between Hoist Rope and Car Frame

Figure 11 shows a comparison of transverse vibration between the hoist rope and car frame. Here, Y3 is the vibration of hoist rope no. 3 in the Y direction, and Y8 is the vibration of the car frame in the Y direction. It is clear from Figure 11 that the vibration amplitude of the hoist rope is 10 times higher that of the car frame. The transverse vibration of hoist rope fluctuates significantly during the initial deceleration stage. This is because the internal tension of the hoist rope is reduced to a certain extent due to the effect of inertia force in this stage under upward movement, and the transverse vibration stiffness is reduced due to the relaxation of the hoist rope. Therefore, the fluctuation amplitude increases in the initial deceleration stage. The peak value of lateral vibration fluctuation of the car frame exhibits a negligible change during the acceleration, uniform speed, and deceleration stages. Because the total gravity of the platform frame and car frame is approximately 12 kN and the transverse vibration is constrained by the guide rail, the vibration amplitude is small and exhibits a minor change.

### 5.2. Time-Frequency Analysis of Transverse Vibration

It can be seen from the analysis of Section 4.2 that the peak value of lateral vibration PSD of the car frame is $3.5\times 10^{-4}m^{2}/s^{3}$. The peak value appears around 12, 29, and 46 Hz. It is evident in Figure 12 that the analysis results of power spectrum are slightly different from those under descending state with no load. The peak PSD increases to $5.7\times 10^{-4}m^{2}/s^{3}$. Moreover, there are only two obvious peaks at 30 and 45 Hz, and no sharp peak is observed at 10 Hz. This implies that the main frequency under downward movement is lower than that under upward movement. This is because even though the elevator is moving upwards and downwards at a constant speed on the macro, when the hoist rope is considered as the research object, due to the constant gravity of the car, the tension of the hoist rope is larger during downward movement, so the main frequency of the transverse vibration of the hoist rope is lower than that during upward movement.

It can be seen from the time-frequency analysis of transverse vibration in Section 4.3.2 that although the main vibration frequencies of the three hoist ropes are different, they all have peaks around 25 Hz. It is evident from the spectrum of Y3 in Figure 12 that the main frequency of the hoist rope is also around 25 Hz under ascending operation without load, and the secondary peaks appear around 16 and 37 Hz. Therefore, it can be inferred that the hoist rope has a peak frequency of 25 Hz at a distance of 1 m from the end of the hoist rope. The peak PSD of vibration is $0.24{\;m}^{2}/s^{3}$, which is 421 times higher than that of the car frame. The lateral vibrations at two different positions differ by two orders of magnitude. Therefore, while controlling the vibration of the wire rope, it is necessary to ensure the smoothness of the guide rail as much as possible.

### 5.3. Comparison of Longitudinal Vibration between Hoist Rope and Car Frame

The comparison of longitudinal vibration between hoist rope and car frame is shown in Figure 13, where Z3 is the vibration of hoist rope no. 3 in the Z direction, and Z7 is the vibration of the car frame in the Z direction. It is clear from the vibration curves that the hoist rope and the car frame vibrate at their frequencies, and the longitudinal vibration frequency of the hoist rope is larger than that of the car frame. The vibration in the acceleration and deceleration stages still represents the coupling between the rigid body movement of the system and the vibration of the components. To analyze the vibration of the components, the longitudinal vibration is filtered in the next section.

### 5.4. Time-Frequency Analysis of Longitudinal Vibration

Similar to the results in Section 4.3.3, the longitudinal vibration of hoist rope no. 3 has energy peaks near 1 and 29 Hz when the elevator is moving upwards (Figure 14). The peak PSD of longitudinal vibration of the hoist rope and car frame is 0.18  and $0.15{\;m}^{2}/s^{3}$ (around 1 Hz), respectively. This is different from the lateral vibration, and there is a minor difference between the peak PSD of the hoist rope and car frame. This indicates that the longitudinal vibration is not sensitive to the slight variation in tension inside the hoist rope. The main energy of longitudinal vibration originates from the energy of system acceleration and deceleration. To clearly analyze the vibration and time-frequency characteristics of the hoist rope during the entire operation, it is necessary to filter the rigid body vibration, which is demonstrated in the next section.

### 5.5. High-Frequency Longitudinal Vibration Characteristics

To clearly analyze the longitudinal vibration characteristics of the hoist rope in the middle- and high-frequency regions, the Butterworth filter is used to filter the vibrations below 1 Hz for removing the influence of rigid body movement on the main frequency of the system vibration. The vibration and time-frequency characteristics are shown in Figure 15 and Figure 16, respectively.

As shown in Figure 15, the longitudinal vibration after filtering is periodic. The longitudinal vibration Z3 of the hoist rope is in the shape of a spindle with two small ends and a large central region, while the longitudinal vibration of the car frame at the initial stages of acceleration and deceleration is slightly higher than that at other regions.

As shown in Figure 16, the main frequency of the hoist rope is nearly 29 Hz, and there is only one obvious peak. Consequently, it can be inferred that the natural frequency of longitudinal vibration of the hoist rope is approximately 29 Hz. The main frequency of the car frame is nearly 4 Hz, and there is a small peak at 29 Hz. This implies that the vibration frequency at the end of the hoist rope is lower than that at the middle region, and the vibrations at the middle and the end regions affect each other.

### 5.6. Comparison and Discussion

Vibration tests are primarily used to examine the safety of the elevator traction system by acquiring the vibration data inside the elevator car. This method is safe, convenient, and easy to implement, but due to the elastic connection between the platform frame and car frame, a large number of vibration signals of the traction system are weakened or cannot be obtained, which leads to ambiguity in the analysis results of vibration data. Based on the comparative analysis considered in this paper, the vibration characteristics of different components can be obtained, and it is believed that the vibration test results of the car frame can reflect the running state of the elevator structure more clearly and accurately, which can provide reference for the design of elevator vibration monitoring sensors and relevant data analysis.

## 6. Conclusions

In this study, an elevator with a traction ratio of 1:1 was used as the experimental platform to analyze the transverse and longitudinal vibration characteristics as well as the time-frequency characteristics of the platform frame, car frame, and hoist rope during upward and downward movement. A test method for examining the coupled transverse-longitudinal vibration of the flexible suspension lifting structure was proposed, and the influence of rope end disturbance on the vibration characteristics of the hoist rope was investigated. The main results of the study are summarized as follows:

(1) The main vibration of the platform frame under normal ascending and descending operation of the elevator was longitudinal vibration. The time-frequency characteristic analysis of the vibration of the platform frame revealed that due to the elastic contact between the platform frame and car frame, the platform frame exhibited low-frequency transverse and longitudinal vibrations, and the main frequency of the vibration in the three directions was basically the same.

(2) The form and amplitude of longitudinal vibration of the car frame were essentially the same as that of the platform frame, but the transverse vibration of the car frame decreased more slowly than that of the platform frame. The time-frequency characteristics indicated that the main frequency of longitudinal vibration of the car was concentrated in the low-frequency region, while the transverse vibration signal exhibited peaks near 10, 30, and 45 Hz.

(3) The comparative analysis of vibration characteristics revealed that the absolute acceleration of the hoist rope during downward movement was lower than that during upward movement due to the gravity of the sensor. Because the hoist rope could only bear tension but not pressure, under the influence of the cage gravity, the tension of the hoist rope during downward movement was relatively larger than that during upward movement.

Overall, this study provides useful insights on designing an elevator monitoring sensor and relevant data processing. Further, it can serve as a useful reference for investigating the vibration characteristics of a vertical suspension system and for analyzing the vibration data of elevator traction systems.

## Figures and Tables

**Figure 1 sensors-20-06586-f001:**
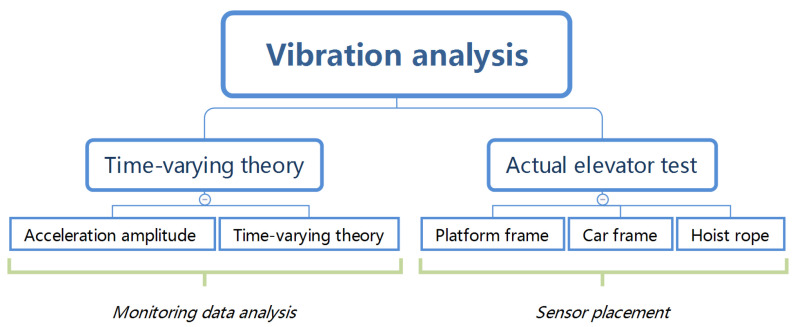
Research workflow.

**Figure 2 sensors-20-06586-f002:**
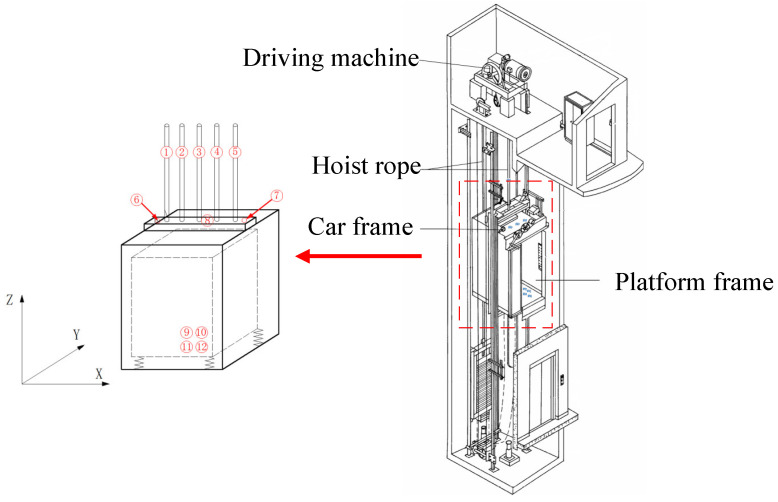
Arrangement and direction of measurement points.

**Figure 3 sensors-20-06586-f003:**
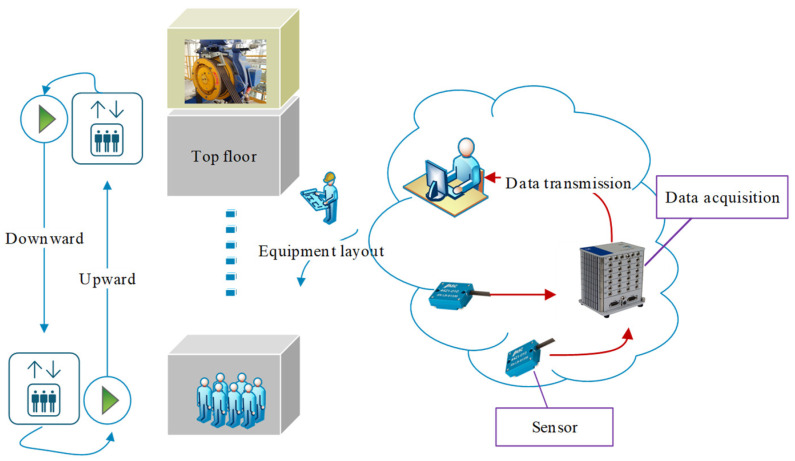
Test process and experimental setup.

**Figure 4 sensors-20-06586-f004:**
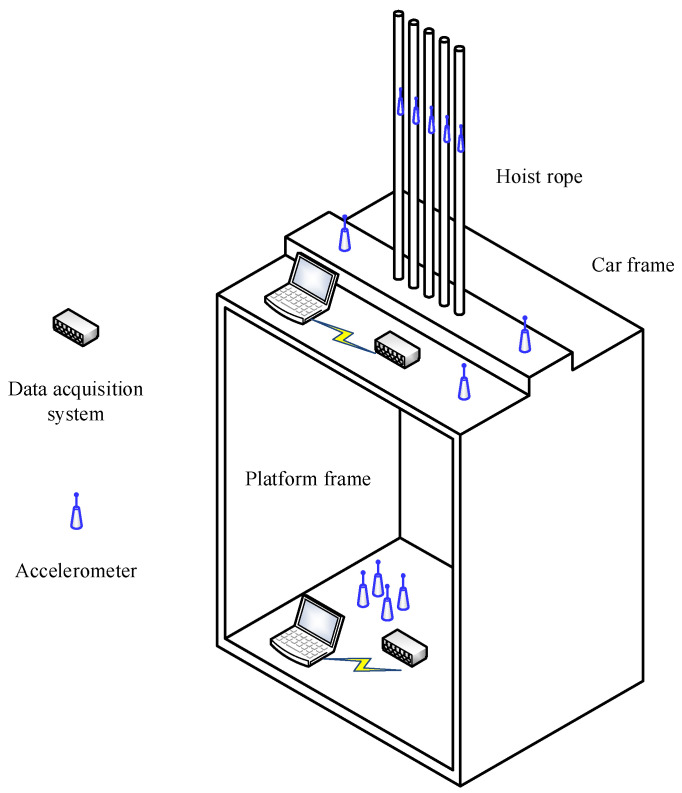
Arrangement of sensors on the platform frame.

**Figure 5 sensors-20-06586-f005:**
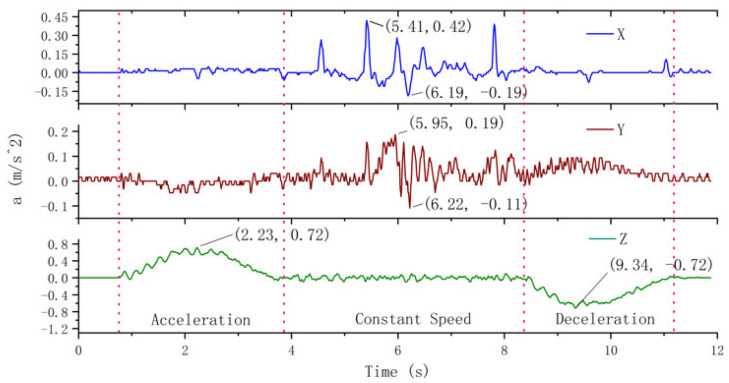
Vibration of the platform frame.

**Figure 6 sensors-20-06586-f006:**
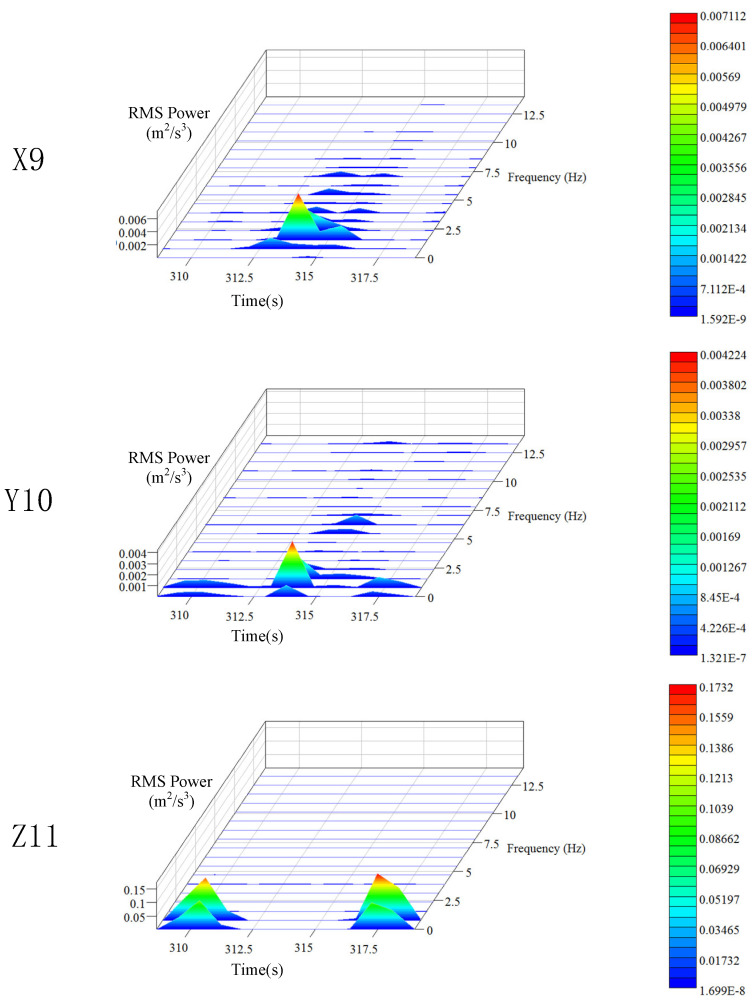
Time-frequency characteristics of the platform frame.

**Figure 7 sensors-20-06586-f007:**
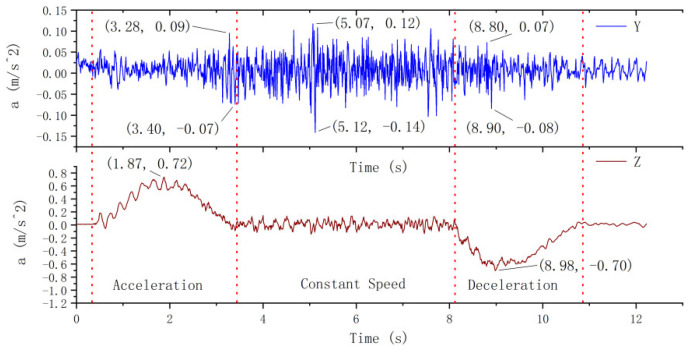
Vibration of the car frame.

**Figure 8 sensors-20-06586-f008:**
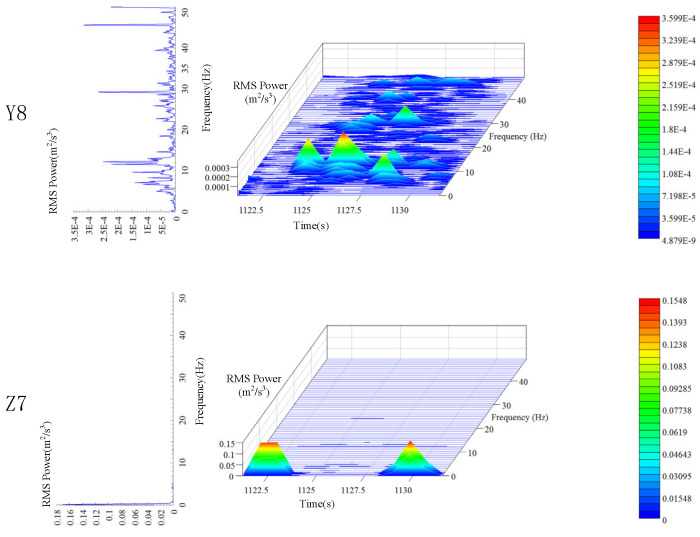
Time-frequency characteristics of car frame.

**Figure 9 sensors-20-06586-f009:**
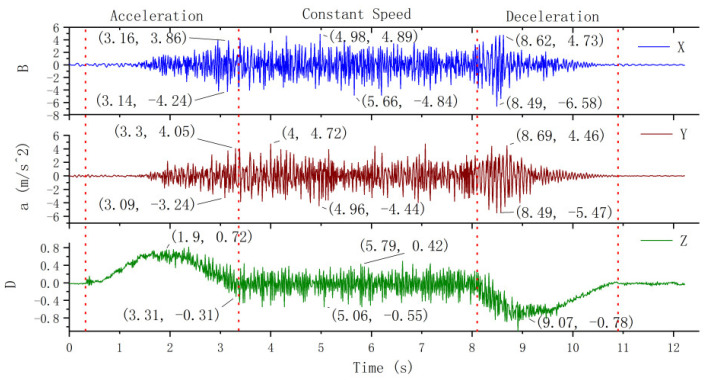
Transverse and longitudinal vibration of hoist rope.

**Figure 10 sensors-20-06586-f010:**
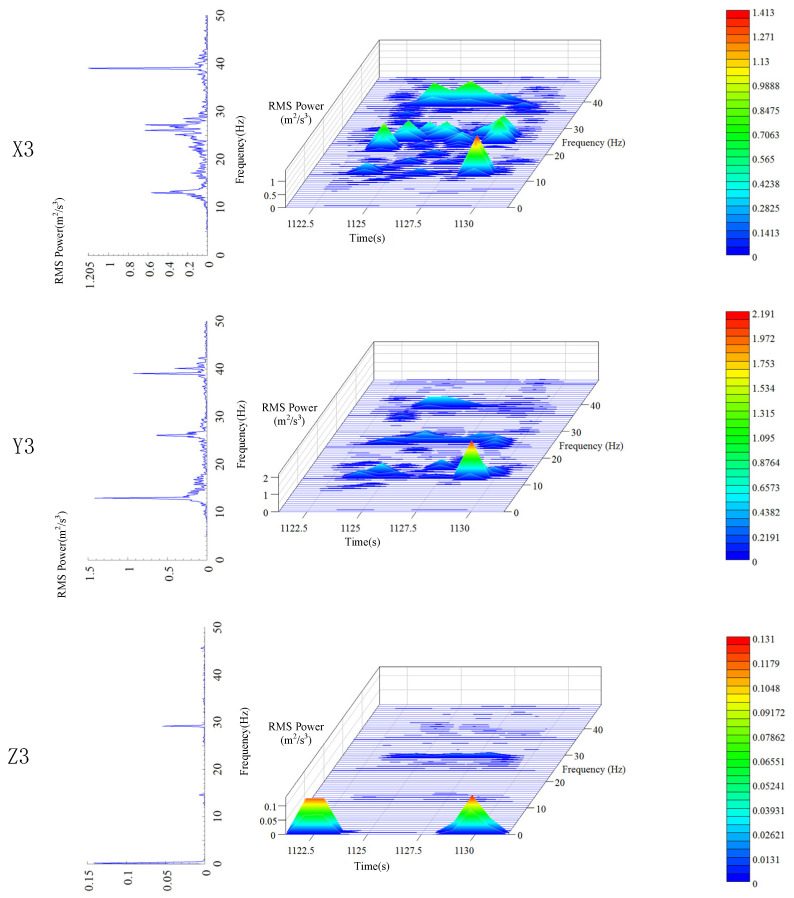
Time-frequency characteristics of hoist rope.

**Figure 11 sensors-20-06586-f011:**
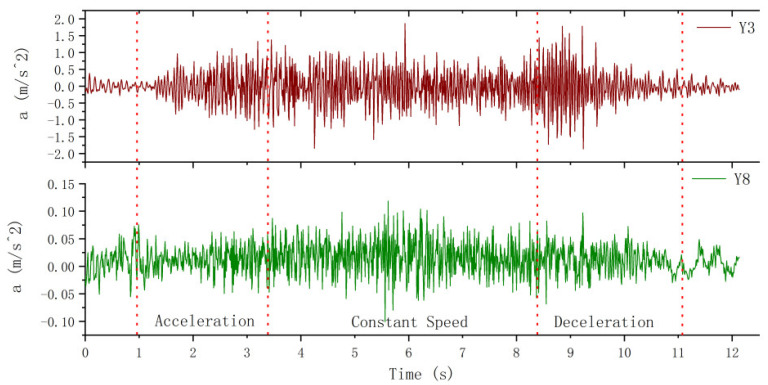
Comparison of transverse vibration between the hoist rope and car frame.

**Figure 12 sensors-20-06586-f012:**
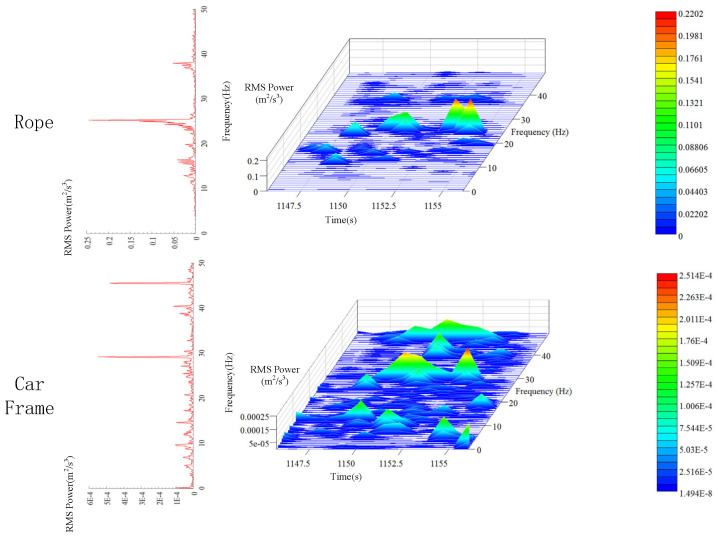
Time-frequency characteristics of transverse vibration of the hoist rope and car frame.

**Figure 13 sensors-20-06586-f013:**
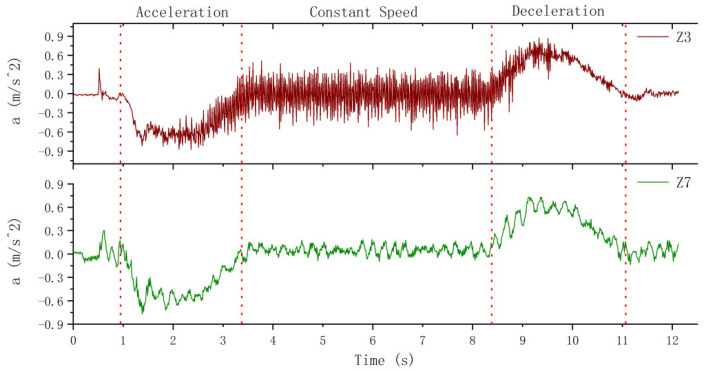
Comparison of longitudinal vibration between the hoist rope and car frame.

**Figure 14 sensors-20-06586-f014:**
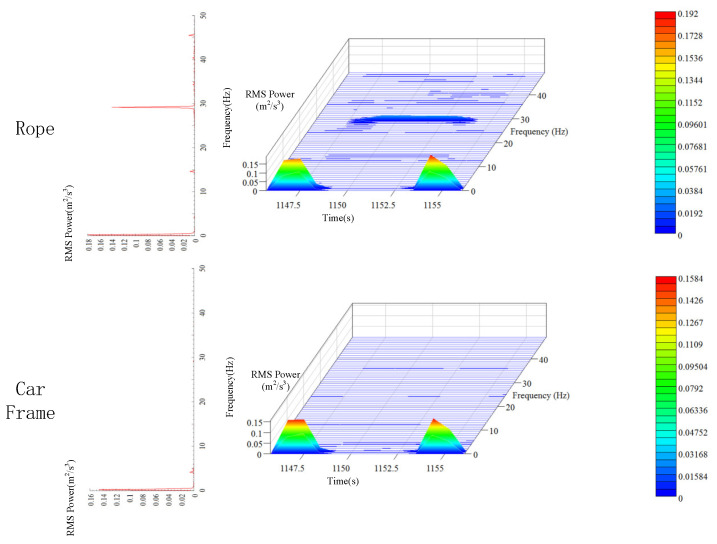
Time-frequency characteristics of the hoist rope and car frame.

**Figure 15 sensors-20-06586-f015:**
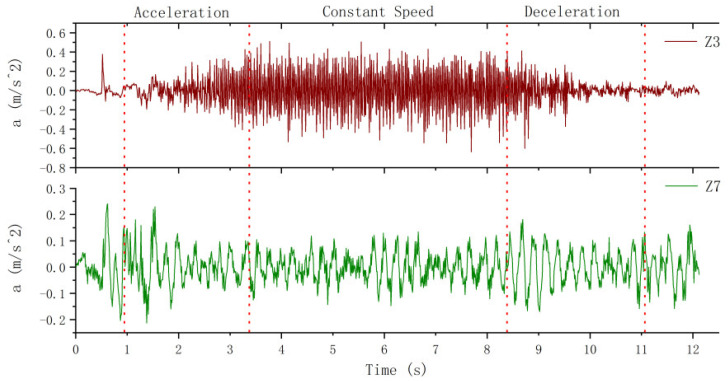
Comparison of longitudinal vibration between the hoist rope and car frame above 1 Hz.

**Figure 16 sensors-20-06586-f016:**
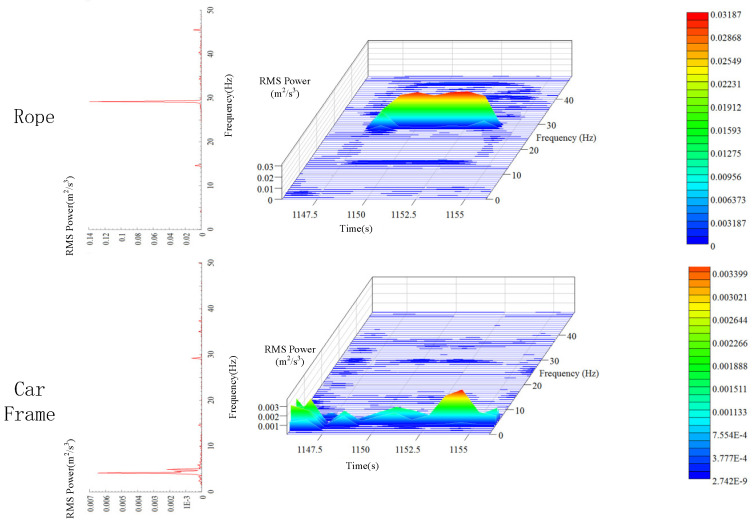
Comparison of time-frequency characteristics between the hoist rope and car frame above 1 Hz.

**Table 1 sensors-20-06586-t001:** Arrangement of measurement points.

Measurement Point	Sensor Number	Position	Direction	Identification Number
1	16-23871	Hoist rope	Z	Z1
2	16-23870	Hoist rope	Y	Y2
3	16-23876	Hoist rope	X, Y, Z	X3, Y3, Z3
4	16-23869	Hoist rope	Z	Z4
5	16-23875	Hoist rope	Y	Y5
6	16-23872	Car roof	Z	Z6
7	16-23874	Car roof	Z	Z7
8	16-23873	Car roof	Y	Y8
9	16-23805	Platform frame	X	X9
10	16-23806	Platform frame	Y	Y10
11	16-23804	Platform frame	Z	Z11
12	16-23808	Platform frame	Z	Z12

**Table 2 sensors-20-06586-t002:** Performance parameters of elevator.

Parameter	Value
Age	10 years
Rated load	1000 kg
Rated speed	1 m/s
Traction ratio	1:1
Quality of car	1218 kg
Quality of counterweight	1638 kg
Hoisting height	7.8 m
Depth of the abyss	1.5 m
Top height	4.2 m
Number of hoist rope	5
Diameter of hoist rope	10 mm

**Table 3 sensors-20-06586-t003:** Parameters of capacitive accelerometer.

Type	Manufacturer	Range	Size(mm^3^)	Material
ASC 4421-001-6A	ASC GmbH	1 g	25.4 × 20.3 × 8	Aluminum

**Table 4 sensors-20-06586-t004:** Peak acceleration of each component in constant speed period ($m/s^{2}$ ).

Measuring Point	X	Y	Z
Car frame	0.42	0.19	0.1
Platform frame		0.14	0.1
Hoist rope	4.89	4.72	0.55

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
