# Peer review of "Analysis of Vibration Monitoring Data of Flexible Suspension Lifting Structure Based on Time-Varying Theory"

_sensors, 2020, doi:10.3390/s20226586_

Round 1
Reviewer 1 Report
This paper proposes a short-time Fourier transform method to investigate the main frequency distribution and influencing factors of the vibration of elevator components. The reviewer has the following questions:
- The innovation of this paper is not obvious. The paper studied the main frequency distribution and influencing factors of the vibration signals of the platform frame, car roof, and hoist rope using the short-time Fourier transform method. There are many relevant researches. The highlights and application value are not enough. The time-varying, operating conditions at high speeds and heavy loads and so on may be research features of the elevator.
- In the conclusions, the authors proposed that based on the analysis of elevator dynamics, the brake fault diagnosis method was studied, which is obviously inconsistent with this paper.
- The elevator system is time-varying, does it mean that its characteristic frequency is variable? Then, whether many characteristic frequencies in the conclusions, such as 1 Hz and 29 Hz, are meaningful? If so, please explain the reasons.
Author Response
Point 1: The innovation of this paper is not obvious. The paper studied the main frequency distribution and influencing factors of the vibration signals of the platform frame, car roof, and hoist rope using the short-time Fourier transform method. There are many relevant researches. The highlights and application value are not enough. The time-varying, operating conditions at high speeds and heavy loads and so on may be research features of the elevator.
Response 1: Thank you very much for the kind comment. The innovation of this paper is that it establishes an effective method for testing and analyzing the running state of elevator structure. The vibration characteristics of elevator platform frame, car frame, and hoist rope are investigated and compared. To highlight the novelty and application value of this study, we have made the following revisions:
- a) The second paragraph of the first section (P.2) is modified to emphasize the innovation and significance of the article.
- b) To clearly explain the research idea, a research flow chart and relevant explanation are added in the last paragraph of Section 2 (P. 4).
- c) Section 4.4 called “Comparison and discussion” is added to compare the peak accelerations of different components in different directions.
- d) The abstract and conclusion sections have been refined to highlight the novelty of the article.
Furthermore, according to your suggestion, we have designed a high-speed, heavy load elevator vibration experiment and will carry out relevant research in the near future.
Point 2: In the conclusions, the authors proposed that based on the analysis of elevator dynamics, the brake fault diagnosis method was studied, which is obviously inconsistent with this paper.
Response 2: We sincerely apologize for this error. This sentence was added by mistake since we were simultaneously writing another article on brake fault diagnosis. Accordingly, this sentence has been deleted.
Point 3: The elevator system is time-varying, does it mean that its characteristic frequency is variable? Then, whether many characteristic frequencies in the conclusions, such as 1 Hz and 29 Hz, are meaningful? If so, please explain the reasons.
Response 3: We appreciate the subtle issue raised by the reviewer. We believe that the characteristic frequency of the overall traction system only fluctuates within a small range due to the large mass of the elevator car. Therefore, the results based on the spectral analysis of the data obtained during single operation of the elevator can reflect the dynamic characteristics of the structure.
The main objective of this study is to analyze the main frequency distribution under different motion states during elevator operation by using time-varying theory. We have been conducting experimental research on examining the relationship between elevator position and frequency distribution and will continue to explore this issue in the future.

Reviewer 2 Report
Dear authors,
your paper intitled "Analysis of vibration monitoring data of flexible suspension lifting structure based on time-varying theory" requires strong revisions. Please check for the following remarks.
- the authors should review the paper by reducing some parts and by focusing on the aim/added value of the proposal. What is the benefit of your study?Did you create a model/map for the vibrations?
- There is no "discussion" section. To my side, the authors should focus the content on the definition of the vibration model. There is no clear trace of this fundamental aspect;
- you submitted the paper on Sensors Jornal. Add a figure with sensors characteristics, locations of sensors, dimensions etc...;
- there are some parts in section 4th that can be summarized or added to supplementary files. Please add a table for improving the readability of the manuscript and for summarizing the data;
Minor revisions:
- # fig. 1: "Driving mechine" --> "Driving machine";
- check for the native English language.
Best regards.
Author Response
Point 1: The authors should review the paper by reducing some parts and by focusing on the aim/added value of the proposal. What is the benefit of your study? Did you create a model/map for the vibrations?
Response 1: We deeply appreciate the suggestion. Accordingly, we have modified Section 4 and the conclusion. To clearly highlight the objective of the study, we have amended the second paragraph of the first section (P. 2). At the end of second section (P. 4), we have described the experimental method and data analysis process. The vibration model was discussed in our previous article "Study on theoretical model and test method of vertical vibration of elevator traction system". This paper mainly focuses on experimental methods and relevant data analysis.
Point 2: There is no "discussion" section. To my side, the authors should focus the content on the definition of the vibration model. There is no clear trace of this fundamental aspect;
Response 2: Thank you for the kind suggestion. Accordingly, we have added a new Section 4.4 called "Comparison and Discussion." Since our main focus is experimental method and data analysis, the vibration model is not described in details.
Point 3: You submitted the paper on Sensors Journal. Add a figure with sensors characteristics, locations of sensors, dimensions etc...;
Response 3: Thank you for the useful suggestion. In the last paragraph of section 3 (P. 6), we have described the sensor characteristics, and the parameters of the sensor are presented in Table 3.
Point 4: There are some parts in section 4th that can be summarized or added to supplementary files. Please add a table for improving the readability of the manuscript and for summarizing the data;
Response 4: Thank you for the kind suggestion. Because the vibration difference between wire ropes is not the focus of this paper, the relevant content in Section 4 has been deleted. To summarize the data, Table 4 (P. 11) is added in Section 4.4 to compare the peak accelerations of different components in different directions.
Point 5: Minor revisions:
- # fig. 1: "Driving mechine" --> "Driving machine";
- check for the native English language.
Response: We sincerely apologize for this mistake. We have corrected the above spelling. Further, we have carefully checked the entire manuscript for native English usage.

Reviewer 3 Report
In this paper, the authors proposed an analysis method of vibration monitoring data of flexible suspension lifting structure based on time-varying theory. The vibration testing methods and also experimental parameters were described well. The research is interesting to the readers. The following comments are provided to the authors for improving this paper.
(1) It is expected that the authors could supplement a flowchart of the proposed method in Section 2.
(2) Please further explain the meaning of the red dot lines in Figure 4 and the other related figures.
(3) Some figures would be much sharper if possible.
(4) The conclusion section should be concise to highlight the novelty of the paper.
Author Response
Point 1: It is expected that the authors could supplement a flowchart of the proposed method in Section 2.
Response 1: Thank you very much for the kind suggestion. Accordingly, we have added a flowchart of the proposed method in the last paragraph of Section 2 (P.4).
Point 2: Please further explain the meaning of the red dot lines in Figure 4 and the other related figures.
Response 2: We appreciate the subtle issue raised by the reviewer. The red dotted lines are used to distinguish the different running states of the elevator, and the relevant explanation has been added in the first paragraph of Section 4.1.1 (P. 6).
Point 3: Some figures would be much sharper if possible.
Response 3: We deeply appreciate the suggestion. Accordingly, we have replaced Figures 3, 6, 9 in the original manuscript by Figure 3 (P.6) in the modified paper.
Point 4: The conclusion section should be concise to highlight the novelty of the paper.
Response 4: We completely agree and appreciate the subtle issue raised by the reviewer. We have refined the conclusion section to highlight the novelty of the article.

Reviewer 4 Report
The authors presented a well-written manuscript. However, I do not see the core novelty of the work. The literature reviewed does not provide the knowledge gap or the motivation. I do not see why these results should be important - this must be argued critically. Several of the results can be guessed qualitatively - on the other hand, the authors have not designed or interpreted the results quantitatively tight either. Under such circumstances I do not see the importance of the work - and nor the novelty. It almost reads like a combination of industrial experiments, which are relevant but without novelty.
I would ask the authors to comprehensively and compellingly argue their case on a) novelty
b) the impact of that novelty (why is it important for the readers)
for all results. this must be carried out in detail. if necessary, please include new analyses.
Author Response
Response: Thank you for the useful suggestion. The innovation of this paper is that it establishes an effective method for testing and analyzing the running state of elevator structure. The vibration characteristics of elevator platform frame, car frame, and hoist rope are investigated and compared. To highlight the novelty and application value of this study, we have made the following revisions:
- The second paragraph of the first section (P.2) is modified to emphasize the innovation and significance of the article.
- To clearly explain the research idea, a research flow chart and relevant explanation are added in the last paragraph of Section 2 (P. 4).
- Section 4.4 called “Comparison and discussion” is added to compare the peak accelerations of different components in different directions.
- The abstract and conclusion sections have been refined to highlight the novelty of the article.

Round 2
Reviewer 1 Report
The authors considered that the innovation of this paper was to put forward an effective method for testing and analyzing the running state of elevator structure. Then, the reason why this method can effectively analyze the running state of elevator system needs to be further clarified. The effectiveness of this method should also be compared with the results of the existing methods for analyzing elevator operation state.
Author Response
Thank you for the useful suggestion. Accordingly, we have clarified the effectiveness of the method in the last paragraph of Section 3 (p.6). Further, we have added a new Section 5.6 (p.15) called "Comparison and Discussion" to compare the proposed method with the existing methods.
Reviewer 2 Report
Dear authors,
your paper intitled "Analysis of vibration monitoring data of flexible suspension lifting structure based on time-varying theory" is now accepted.
Please, I'm suggesting to use red color for the revisions on the paper for the future. The readability of the "blue" revisions is not so detectable with respect to the "black" .
Best regards.
Author Response
Thank you very much for the kind suggestion. Accordingly, the second revisions in the paper have been marked in red.
Reviewer 4 Report
The paper can be accepted from my side
Author Response
Thank you for accepting our article.